# Power and influence in world-level sport coaching: A Foucauldian and Raven-informed interpretive vignette study in underwater rugby

Samuel José Gaviria-Alzate[ID][1]*, Claudia Arcila-Rojas[2], Luis Álvarez González[3], Juliana Posada Monsalve[3], Jonnathan Arenas Taborda[3], Lina Restrepo[3], Wilder Geovanny Valencia-Sánchez[ID][4]

1 Education and Social Science, Tecnológico de Antioquia Institución Universitaria, Medellín, Colombia, 2 Universidad de Antioquia, Medellín, Colombia, 3 Education, Universidad de San Buenaventura Medellín, Medellín, Colombia, 4 Instituto Universitario de Educación Física, Universidad de Antioquia UdeA Medellín, Colombia

* samuel.gaviria@tdea.edu.co

## Abstract

This study examined how world-level underwater rugby coaches mobilise interpersonal power to shape athletes' behaviour, cohesion and self-regulation under high-pressure, low-verbal conditions. Anchored in Foucauldian accounts of power/knowledge and Raven's interpersonal power bases, we examined how trust, expertise and authority were narrated within an embodied coaching ecology. An interpretive qualitative vignette design with a hermeneutic sensibility was employed with three world-champion coaches (aged 42–44; ≥ 12 years' international experience). Data were generated via an online questionnaire comprising ten high stakes coaching scenarios. For each vignette, participants selected the single response option that best approximated their likely first-line action (mapped at design time to French and Raven's power bases, including informational power) and provided a brief open-text rationale. Analysis combined a deliberately modest descriptive use of the closed selections with reflexive thematic analysis of the written rationales, treating the taxonomy as a sensitising framework at interpretation. Three themes captured coaches' influence work: relational anchoring under aquatic risk; expertise as embodied pedagogy, in which brief reasons were folded into demonstration; and boundary-setting governance, in which formal authority was framed as selective and proportionate. The contribution is not a general revision of power theory, but a context-specific refinement. In low-verbal aquatic settings, influence was narrated as a compressed choreography in which referent trust opens receptivity to expert demonstration, informational reasoning is nested within expert guidance, and formal authority operates mainly as a bounded safety and standards resource.

**Data availability statement:** The de-identified closed-choice selections underlying Table 3 are provided as Supporting Information (S1 Dataset, CSV). A scenario-by-scenario descriptive summary of closed selections is provided as Supporting Information (S1 Table). The vignette prompts and decision-task structure are provided as Supporting Information (S2 Appendix), alongside an analytic codebook (S3 Appendix) and a reflexive memo documenting analytic decisions (S4 Appendix). To protect participant confidentiality given the small, readily identifiable elite population, the full open-text rationales are not publicly shared. Qualified researchers may request access to a further-redacted version of the qualitative materials from the corresponding author, subject to ethical considerations and an appropriate data-sharing agreement.

**Funding:** The author(s) received no specific funding for this work.

**Competing interests:** The authors have declared that no competing interests exist.

## Introduction

The efficacy of athlete development and collective success in high-performance sport is inseparable from the quality of coach-athlete relations [1,2]. Coaching is not merely technical instruction; it is a dynamic, relational practice embedded in influence, meaning making, and governance [3,4]. A substantial body of work has examined how athletes interpret coaches' influence, including studies that link perceived power bases to satisfaction, commitment and communicative evaluations in team sport settings [5,6]. However, empirical knowledge remains uneven regarding how influence is mobilised when communication channels are structurally constrained and performance depends on rapid, embodied coordination.

Underwater rugby provides a distinctive, high-pressure ecology for examining influence [7,8]. Immersion restricts verbal exchange, compresses decision time, and increases reliance on kinaesthetic, tactile and gestural cues for synchrony and safety [9,10]. In such low-verbal environments, influence must often be enacted through presence, timing, exemplarity and brief pre-breath 'micro-briefings', rather than extended explanation [11,12]. Despite growing interest in elite coaching and leadership, there is limited empirical work that specifies how interpersonal power is enacted in underwater or similarly bandwidth-constrained sport contexts, and how classic power taxonomies translate to these conditions [13,14].

To interrogate this problem with conceptual precision, the present study aligns two complementary lenses selected for their analytic fit at different scales. Foucauldian analysis frames coaching as situated within an apparatus of power/knowledge in which training practices, evaluation routines and performance discourses jointly shape what athletes come to recognise as 'good' action and 'acceptable' conduct [15,16]. In parallel, French and Raven's typology specifies the micro-foundations of interpersonal influence through legitimate, reward, coercive, expert and referent power, later expanded by Raven to include informational power [17,18]. Together, these lenses allow us to examine both the broader governance of conduct and the interactional influence bases coaches prioritise in concrete dilemmas. We therefore use the taxonomy not as a fixed codebook, but as a sensitising framework for interpreting how coaches describe influence in practice.

Accordingly, we asked: (1) How do world-level underwater rugby coaches describe influencing athletes in recurrent high-stakes situations under low-verbal constraints? (2) Which bases of interpersonal power are foregrounded when coaches select their likely actions across scenarios, and how do they justify those choices? and (3) How do coaches' written rationales indicate blending, sequencing or bounded use of formal authority in an aquatic risk environment? By combining vignette-based elicitation with reflexive thematic analysis of coaches' rationales, we aim to make a bounded theoretical move. Rather than proposing a universal revision of French and Raven's taxonomy or a general pedagogy of elite coaching, we argue that in bandwidth-constrained coaching ecologies the bases may be narrated in compressed, practically entangled form. Referent and expert power become the dominant actionable pair, informational influence is often carried within expert demonstration, and legitimate authority is retained mainly as a situational backstop for safety and standards.

## Methodology

### Design and philosophical stance

An interpretive qualitative design with a hermeneutic sensibility [19] was employed to examine how influence is understood and enacted in elite underwater rugby coaching. The Foucauldian and French-Raven frameworks were selected because they operate at complementary analytic scales. The former helps interpret how conduct is normalised, monitored and ethically governed in context, whereas the latter provides a mid-range vocabulary for the influence levers visible in scenario responses. Analytically, reflexive thematic analysis (RTA) [20,21] was conducted with a predominantly inductive logic, while French and Raven's power bases (including informational power) were used as sensitising concepts at interpretation rather than as a priori coding categories [18,22]. Initial coding proceeded openly at semantic and latent levels; reflexive memos documented how the data resonated with, stretched or troubled the taxonomy; and theme development was iteratively refined within the hermeneutic circle to preserve participants' meaning making. A second analyst acted as a critical friend to interrogate theme boundaries and evidential adequacy; consistent with RTA, no consensus coding or inter-rater reliability was pursued because analytic rigour was sought through reflexivity, transparency and critical interrogation rather than coder agreement [23].

### Setting and context

The aquatic milieu constrains verbal exchange and heightens kinaesthetic coordination, making underwater rugby a theoretically apt site for examining how coaches narrate influence when timing, embodiment and safety compress communicative bandwidth.

### Participants

Purposive, criterion-based sampling [24,25] recruited three male coaches (aged 42–44), all current or former world champions and head coaches of their national teams with ≥12 years' elite coaching experience. Two had also served as active player-coaches during parts of their careers. This was a deliberately narrow, information-rich sample rather than an attempt at representativeness. Sample adequacy was judged in relation to the specificity of the research question, the rarity of the population, and the density of elicitation generated by ten scenarios per participant, rather than by code saturation across a heterogeneous field. In other words, we did not pursue saturation as a formal endpoint; instead, we sought analytically adequate, high-expertise accounts capable of illuminating patterned influence logics within a tightly bounded elite context [24,25,26,27]. To contextualise the sample while preserving anonymity, brief profiles are provided. Coach 1 (highest championship record) had extensive international tenure, completed doctoral training in sport, and spent three seasons competing in European club championships. Coach 2 reported long-standing international coaching and postgraduate training in sport. Coach 3 also reported long-standing international coaching and professional qualifications in coaching. All three held foundational degrees in sport-related fields. These characteristics were not used as a priori analytic categories; rather, they provided contextual background that informed cross-case interpretation during theme refinement.

### Scenario design

To prompt practical reasoning while preserving ecological validity, a scenario-based elicitation guide comprising ten recurrent, high-stakes coaching situations was developed using vignette methodology [28]. Each vignette portrayed an actionable dilemma typical of elite underwater rugby (e.g., authority challenges, post-defeat demoralisation, mid-game disengagement). For instrument design purposes only, the research team pre-identified the power base(s) most plausibly implicated (Primary) and those commonly co-activated (Secondary), using French and Raven's taxonomy (including informational power) as a heuristic for generating plausible response options [18,22]. Each vignette required a single

forced-choice response. We adopted a single-option format to compel prioritisation of the action coaches judged most likely to take first under time-pressured conditions, while the accompanying open-text rationale was used to recover nuance, blending and exceptions. The instrument was conceived as an elicitation device rather than a psychometric scale; accordingly, we sought contextual realism, face validity and discriminability of options through iterative expert review, rather than internal consistency or reliability coefficients. We acknowledge that the design-time mapping may prime theoretically expected interpretations; this risk was mitigated by (i) presenting the task as a decision-making exercise without highlighting power terminology, (ii) collecting an open-text rationale after each single choice, and (iii) conducting inductive coding of rationales independently of the design-time map, which was revisited only at interpretation to examine convergence, divergence and ambiguity. The full vignette wording and response structure are provided in S1 Appendix.

## Data collection, management, analysis

Data were generated in a single phase via an online vignette questionnaire delivered through a secure web form. After reading an electronic participant information sheet, all respondents provided written e-consent (checkbox acknowledgement) before proceeding. Recruitment ran from [02 February 2025] to [16 February 2025], and each participant completed the questionnaire once within that window. A brief framing note introduced the general topic (influence and decision-making in high-performance environments) without disclosing specific hypotheses or the intended analytical focus on power.

The questionnaire comprised ten scenarios representing recurrent, high stakes coaching situations in elite underwater rugby (Table 1). For each vignette, participants selected the single response option that best reflected their likely first-line action; options were mapped, at design time, to French and Raven's bases, including informational power, and were followed by a short open explanation in the coach's own words. The instrument was designed by two authors with complementary expertise in qualitative methods and elite coaching (>10 years each). Content clarity, realism and option differentiation were enhanced through expert review.

Structured outputs (multiple-choice selections) and open-text justifications were exported from the form (CSV and UTF-8 text), anonymised, and imported into NVivo for secure storage, retrieval and code management [29]. The anonymised closed-choice selection dataset underlying the descriptive counts is provided in S1 Dataset. Pseudonyms (Coach 1/2/3) were assigned at import, and an audit trail recorded versioning of data, analytic memos and theme maps.

Two complementary tracks were undertaken.

Table 1. Scenario-power base map and analytic cues for vignette-based elicitation.

| # | Scenario focus | Primary base(s) | Secondary base(s) | Analytic cue |
|---|---|---|---|---|
| 1 | Authority challenge | Legitimate, referent | Informational | Authority vs. identification |
| 2 | Demoralisation post-loss | Referent | Expert | Restoring efficacy/care |
| 3 | Dyadic conflict | Referent | Legitimate | Mediation, boundaries |
| 4 | Communicating new strategy | Informational, expert | Referent | Scaffolding/demonstration |
| 5 | Underperforming key player | Expert | Referent | Feedback vs. trust |
| 6 | Checking understanding | Informational | Legitimate | Surveillance-as-care |
| 7 | Mid-game disengagement | Referent | Legitimate | Presence/tempo |
| 8 | Complacency when leading | Legitimate | Referent | Standards vs. inspiration |
| 9 | Frustration after comeback | Referent | Informational | Emotional regulation |
| 10 | Integrating newcomers | Referent | Legitimate | Identity/inclusion |

*Note*. "Primary" denotes the design-time judgement of the most likely power base(s) implicated by the scenario; "Secondary" indicates bases expected to co-activate in practice. Assignments are not mutually exclusive and were used solely to structure prompts and follow-up probes; they were not used as a codebook nor for statistical inference. "Informational" refers to reason-giving/evidence-based persuasion; "Legitimate" refers to role, or norm-based authority. "Analytic cue" summarises the practical judgement the vignette was intended to elicit, guiding probing while leaving coaches' accounts open.

1. Descriptive depiction of selections. Closed-choice selections were tallied per scenario and per coach and are reported as descriptive counts only (no statistical inference). Their role was not to evidence prevalence or effectiveness, but to preserve what coaches foregrounded when compelled to prioritise a single first-line action before elaborating their reasoning. Read alongside the rationales, the tallies provide an orienting layer that makes visible where subsequent explanations confirmed, qualified or effectively expanded the prioritised choice.

2. Reflexive thematic analysis (RTA). The written rationales accompanying each selection were analysed using RTA [20,21,23]. Analytic work unfolded in six recursive steps: (1) repeated familiarisation with all responses; (2) open coding of meaning units without applying the power-base taxonomy; (3) memo-writing to capture candidate patterns, tensions and absences; (4) clustering of related codes into provisional interpretive groupings; (5) iterative development, testing and renaming of themes against the full dataset and scenario context; and (6) a final interpretive pass relating the themes back to the descriptive selections and the sensitising frameworks. Codes were short action-oriented labels (for example, trust restoration through presence, concise cue plus reason, or authority reserved for non-negotiable safety), which were later assembled into broader thematic patterns. A structured summary of the analytic codebook, including code labels, descriptions, and theme-development organisation, is provided in S1 Analytic Codebook. The power-base taxonomy and Foucauldian ideas operated strictly as sensitising concepts at interpretation; initial coding remained open and data-led, and reflexive memos tracked analytic decisions and alternative readings.

Given the brevity of the written rationales and the small, deductively identifiable elite sample, results are reported through closely excerpt-linked analytic description with pseudonyms (Coach 1/2/3) and scenario identifiers where relevant, rather than through numerous standalone quotations.

### Ethics approval and consent to participate

**Ethical considerations.** Procedures adhered to accepted standards for research with human participants and to the most recent revision of the World Medical Association's Declaration of Helsinki (2024). Ethical approval was granted by the Institutional Research Ethics Committee of the local university, approved [12-12-2024; Act No. 10]. All participants provided electronic informed consent prior to data collection. Confidentiality safeguards (pseudonymisation, removal of indirect identifiers, and aggregated reporting of frequencies) were implemented to minimise risks of deductive disclosure in this small, elite population [30,31].

### Trustworthiness and quality

Credibility and transparency were supported by thick description, an explicit audit trail (protocol versions, analytic memos and evolving theme maps), and structured critical-friend debriefs [26,27]. Given the questionnaire-based design, we followed SRQR guidance and reported COREQ items where applicable to enhance completeness [32,33] (Table 2).

## Results and discussion

The study examined how world-level underwater rugby coaches mobilise interpersonal power to influence athletes' behaviour, cohesion and self-regulation under high-pressure conditions.

Across the 30 closed-choice responses (10 scenarios per coach), coaches selected referent power 19 times and expert power 11 times; informational, legitimate, reward and coercive bases were not selected in the closed options (Table 3). On their own, these counts are analytically modest. Their value lies in showing what coaches prioritised when forced to nominate a single first-line action, which can then be read against the rationales to identify convergence, qualification and hidden co-activation. Accordingly, the tallies are presented only as an orienting descriptive layer and should not be interpreted as prevalence, effectiveness or normative desirability.

**Table 2. Trustworthiness strategies.**

| Criterion | Procedures enacted | Materials retained |
|---|---|---|
| Credibility | Thick description; exemplar quotations; critical friend debriefs; active search for disconfirming cues during analysis | Memo log; excerpt index with scenario IDs |
| Dependability | Reflexive journal; traceability from excerpt→code→theme→claim | NVivo exports; dated analytic artefacts; protocol versions |
| Confirmability | Reflexive journal; traceability excerpt claim | Journal entries; code–theme traceability matrix |
| Transferability | Rich sport context; sampling rationale; scenario prompts and decision-task structure provided in S1 Appendix | Methods text; Table 1; S1 Appendix |

Note. Criteria follow established qualitative quality guidance. "Critical friend debriefs" = structured peer discussion to surface assumptions and test alternative readings. The audit trail comprised protocol versions, reflexive memos and evolving code structures/theme maps. Quotations derive from written rationales submitted in the online questionnaire.

**Table 3. Frequency of selected power bases by coach (10 scenarios).**

| Coach | Referent | Expert | Informational | Legitimate | Reward | Coercive | Total |
|---|---|---|---|---|---|---|---|
| C1 | 8 | 2 | 0 | 0 | 0 | 0 | 10 |
| C2 | 5 | 5 | 0 | 0 | 0 | 0 | 10 |
| C3 | 6 | 4 | 0 | 0 | 0 | 0 | 10 |

Note. Frequencies derive from single forced-choice selections linked to French-Raven bases for each vignette; values are descriptive only and are intended as an orienting layer for reading the open rationales.

### Thematic findings from open justifications

Analysis of the open-text justifications accompanying each selection produced three themes that specify how influence was reported as enacted within an embodied, low-verbal aquatic ecology. Traceability was strengthened by reading each theme back to the scenario types that most consistently elicited it. Trust-based recalibration around demoralisation, disengagement and integration; competence-led adjustment in strategy and performance-correction scenarios; and bounded authority in dilemmas involving standards, role clarity and safety.

For analytic clarity, themes are presented separately; however, the accounts indicate dynamic co-activation. Referent trust was described as establishing the relational conditions under which expert guidance could be received, while formal authority appeared primarily as boundary-setting around safety and standards rather than as a dominant influence strategy. We actively searched for counter-narratives, tensions and ethically problematic influence practices. The vignette format and the reputational stakes of a world-level sample likely encouraged normatively "polished" accounts, so absence of overt conflict in these data should not be read as evidence that conflict is absent in practice.

**Theme 1. Embodied trust under aquatic risk (referent power).** Influence was frequently grounded in identification and trust, enacted through presence, shared rhythm and exemplariness rather than overt direction. This pattern was especially visible in scenarios involving demoralisation after loss, mid-game disengagement, frustration after a comeback and the integration of newcomers. Coaches described low-verbal coordination and timing as central to securing alignment, with role-modelling and relationship quality linked to players' uptake of plans. This 'doing with' rather than 'telling' stance was salient when addressing setback and re-stabilisation demands (cf. Fig 1; Table 3).

**Theme 2. Expertise-as-demonstration (expert/informational reasoning).** In scenarios requiring technical recalibration, especially communicating a new strategy or addressing an underperforming key player, coaches foregrounded competent diagnosis and concise demonstration to anchor execution. Although informational did not appear

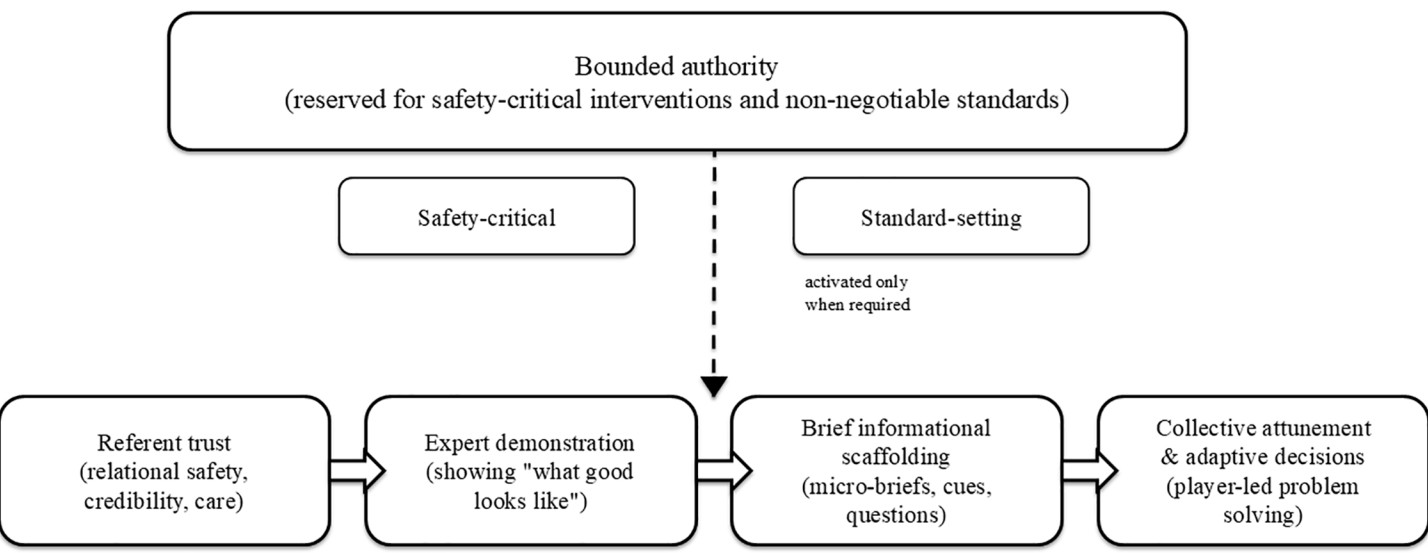

**Fig 1. Conceptual model of low-bandwidth influence choreography in elite underwater rugby coaching.** Note. The figure is intended as a heuristic synthesis rather than a sequential, exhaustive or prescriptive model. It summarises how the themes relate under low-verbal aquatic conditions by illustrating how referent trust can establish relational conditions for expert demonstration and brief informational scaffolding, while bounded authority remains available for safety-critical and standard-setting moments.

in the closed tallies, the rationales repeatedly invoked brief why-explanations, tactical clarification and evidence-like justification embedded within concise, expert-led guidance. This suggests that informational work was not absent so much as pragmatically folded into expert delivery in the moment (see Fig 1).

**Theme 3. Calibrating authority (selective, bounded recourse to formal power).** References to legitimate, reward or coercive bases were minimal in the written accounts. When raised, authority and consequences were framed as situational and bounded, especially in scenarios concerning authority challenge, checking understanding, complacency while leading, or other moments where standards and safety became non-negotiable. Checks for understanding were positioned as supportive monitoring consistent with trust-preserving practice, rather than surveillance for compliance.

Taken together, Table 3 and the qualitative explanations converge on a pattern in which referent and expert bases dominate coaches' forced prioritisation, while the rationales reveal how informational reasoning is nested within expert practice and how formal authority is held in reserve rather than foregrounded.

## Cross-case patterns with participant-specific nuance

Descriptive closed-choice selections are reported as counts only (S1 Table) to avoid false precision with n = 3; the qualitative themes are grounded in the written rationales.

**Coach 1 (8 referent / 2 expert).** Influence was described as attunement-first; re-establishing tempo through presence, eye contact and brief signals before issuing minimal, high-leverage cues. Authority was framed as earned via consistency and exemplariness; standard-setting and recognition displaced overt rewards or sanctions. In setback scenarios, shared rhythm and role-model conduct preceded technical adjustment, an ordering in which referent power legitimised succinct expert direction (Fig 1).

**Coach 2 (5 / 5).** Accounts indicated purposeful toggling between relational scaffolding and technical instruction. Brief "why" explanations were paired with concrete cues, evidencing informational reasoning embedded within expert delivery.

References to legitimate authority surfaced mainly to clarify roles or mediate dyadic frictions and were quickly re-absorbed into trust-preserving dialogue. The profile reads as a balanced constellation responsive to situational demands.

**Coach 3 (6 / 4).** Narratives conveyed directive brevity under time pressure, with rapid, specific corrections and readiness to demonstrate key actions. Simultaneously, the coach reported using micro-validations (brief affirmations, eye contact) to protect autonomy and relatedness. Formal authority was bounded, reserved for safety and non-negotiable standards, mirroring the broader tendency to minimise coercion while leaning on competence-based credibility.

Across participants, influence appears as a situated blending of referent and expert bases, with informational work woven into the latter and formal authority calibrated to preserve trust. The subjective variation maps onto stylistic preferences under pressure, from trust-first sequencing (Coach 1), through balanced toggling (Coach 2), to directive brevity with autonomy safeguards (Coach 3), offering a context-specific specification of how power/knowledge operates in underwater, low-verbal coaching ecologies.

## Referent-expert complementarity under aquatic constraints

The contribution of these findings is intentionally bounded. We do not claim a general revision of French and Raven's taxonomy; rather, the data support a context-specific refinement for low-bandwidth coaching ecologies. In this setting, influence appears not as a loose accumulation of separate bases but as a compressed choreography in which referent trust prepares receptivity, expert power carries the actionable adjustment, informational influence is often nested within the expert cue, and legitimate authority remains a reserve resource for safety and standards. The predominance of referent and expert power therefore suggests that influence is scaffolded by the synergy of identification and competence. Under breath-hold cycles and constrained verbal bandwidth, coaches reported relying on presence, timing and example to secure buy-in, while concise technical framing anchored execution. Such accounts resonate with a Foucauldian view of power/knowledge, with authority reproduced through practice, examination and normalisation rather than imposed externally [15,16], and with Arendt's distinction between power (acting in concert) and violence (force), implying that durable authority in champion teams rests on collective trust rather than coercion [34]. From a sociological angle, Bourdieu's habitus and symbolic capital help explain why demonstrated mastery and reputational histories legitimised coaches' classifications and decisions [35].

At the micro-interactional level, the findings refine French and Raven's taxonomy for this context by showing how informational and expert bases converge in practice; reason-giving appears most workable when coupled with demonstrable know-how, particularly where verbal exchange is limited [18,22]. One coach exhibited a balanced referent-expert composition (50/50; Coach 2), while the others leaned more referent, suggesting that sustained effectiveness may hinge less on the volume of a single base than on the composition and timing of bases across situations [2–4]. This is also why referent and expert power likely emerged so prominently. Both are actionable under compressed communicative conditions, whereas reward, coercive and even legitimate strategies require either more explicit negotiation or a more overt enforcement climate than coaches reported as functional or desirable here.

**'Silent leadership', subjectivation and ethical government.** Phenomenological descriptions of underwater coordination, tempo, touch, breath, support the idea that meaning is enacted by the played body [36,37]. Within a Foucauldian frame, these mundane practices function as technologies of the self through which athletes learn to govern themselves by shared norms of timing, effort and attention [15,38]. Coaches' preference for recognition, exemplarity and supportive monitoring over punitive control aligns with governmentality rather than domination: the conduct of conduct exercised ethically to produce reliable action under pressure [4,39]. Where formal authority appeared, it was bounded by role standards and articulated as a situational safeguard, consistent with autonomy-supportive coaching and established motivational science [40,41].

There are contextual reasons why this constellation likely emerged. Underwater rugby is an embodied, low-verbal, risk-aware milieu; breath-hold cycles [42], limited talk time and aquatic hazards privilege presence, timing and demonstration

over extended verbal persuasion, making a referent-expert sequence both efficient and acceptable. The elite profile of the participants also matters; all were long-tenure national coaches with advanced training in sport (and, for one, doctoral preparation) and substantial international exposure [43,44]. In parallel, it is common for athletes in this discipline to have comparatively strong academic backgrounds, which plausibly supports reflective practice, collegial problem-solving and conflict de-escalation via shared professional norms rather than via formal sanctions [45,46]. Moreover, unlike stadium sports, the absence of intense crowd pressure and a smaller media ecosystem reduce externally amplified stressors; coaches can rely on relational trust and technical clarity rather than on public displays of authority [47,48].

## Alternative explanations and boundary conditions

Several alternative explanations merit explicit consideration. First, elite coach professionalisation may account for the predominance of autonomy-supportive influence: world-level coaches are often socialised into contemporary pedagogical norms that privilege athlete self-regulation, which could produce convergent accounts independent of the aquatic ecology [14,49,50]. Second, cultural norms within underwater rugby, including long-term team continuity and shared national per-formance narratives, may amplify relational identification and thereby increase the salience of referent power [51]. Third, the questionnaire format may elicit idealised "best practice" responses; the absence of conflict or problematic influence should therefore be interpreted as a function of social desirability and self-presentation, not as evidence that resistance or coercion are absent in practice [52,53].

Boundary conditions are equally important. The findings should not be assumed to generalise to youth sport, commer-cial coaching, or high-turnover squads, where legitimate and reward-based influence may play a larger role. Moreover, the forced single-choice design compresses co-activation and sequencing; in vivo influence is likely to involve rapid switching between relational calibration (referent), brief reason-giving (informational), and demonstration (expert), with bounded authority invoked primarily for safety-critical or standard-setting moments.

The group-management consequences are direct. A referent-first approach appears to secure buy-in and cohesion rap-idly under pressure, while subsequent expert framing focuses collective attention on high-leverage adjustments. Together, these mechanisms may stabilise self-regulation and reduce the need for coercive resets [54,55]. In this sample, the coach with the strongest championship record exhibited the most balanced referent expert profile, suggesting, cautiously, that composition and timing of bases, rather than the sheer volume of any single base, may underpin reliable execution in critical moments [56,57]. Although causal claims cannot be made from a small, questionnaire-based design, the present analysis specifies how power/knowledge is enacted in underwater, low-verbal settings and why this particular blend is functional there, reinforcing the case for cultivating identification building and trust maintenance as foundational skills, pairing them with concise, competence-led "micro-rationales", and reserving formal authority for clearly justified contingen-cies so that autonomy and cohesion are preserved while standards remain non-negotiable [58,59].

## Methodological contribution

Methodologically, combining a scenario-based vignette questionnaire with reflexive thematic analysis (RTA) of the open-text justifications enabled a dialogue with the French-Raven bases that neither a purely deductive codebook nor frequency counts alone could provide [20,21,23,28,60]. The single forced-choice selections yielded a constrained compositional map of what coaches prioritised first (Table 3; Fig 1), while the open responses exposed the mechanisms, sequencing and thresholds by which influence bases were narrated in practice (e.g., trust-first alignment followed by concise technical direction). Treating the taxonomy as sensitising concepts, rather than fixed codes, allowed the analysis to register co-activation, ordering and conditions for switching, all within a single, tightly bounded design.

The approach also clarifies apparent absences in the tallies. Although informational did not appear as a closed selec-tion, the open justifications consistently embedded reason-giving within expert delivery (brief "why" statements coupled to precise cues), indicating subsumption rather than lack. Likewise, the rarity of legitimate, reward and coercive selections

is contextualised by bounded-use rules that surfaced in text (e.g., safety enforcement, norm breaches), specifying when formal authority becomes acceptable and why it is usually minimised. Thus, the vignette-plus-RTA design recovers latent structure in coaches' influence work (blends, sequences, safeguards) that a frequency-only approach would miss, while avoiding trait-like reification from a rigid deductive codebook [18,22].

## Practical implications for coach education

Practical implications should be read in two layers. The direct implications apply to underwater rugby and comparable low-verbal, risk-aware performance environments in which influence must often be enacted pre-emptively and embodied. Any broader relevance should be treated as a transferable hypothesis for future comparative work, not as an evidence-based prescription for coaching in general.

Second, the prominence of expertise-as-demonstration suggests that instructional content should prioritise low-bandwidth teaching tools over extended verbal explanation. Coaches can be trained to convert complex tactics into short pre-breath scripts (one or two cue words), to use poolside demonstration and constrained rehearsal to encode timing, and to pair any directive with an immediate, concrete rationale (why this, now) that fits within the available communication window. Video-assisted debriefs after sets or sessions (freeze-frame, short clips, athlete self-explanation) can extend informational power on land, freeing underwater communication for execution cues.

Third, bounded authority in this ecology appears most defensible when it is explicitly safety-anchored and procedurally transparent. Coach education should therefore include protocols for safety-critical governance (e.g., breath management norms, substitution rules, contact thresholds, and 'stop-play' signals) that are co-developed with athletes and applied consistently. Where sanctions or consequences are necessary, they should be tied to clear standards and followed by rapid relational repair, preserving the referent conditions that make future expert guidance effective.

Finally, coach developers in comparable constrained-communication environments may find vignette-based exercises similar to those presented here useful as reflective training devices, especially when prompts force consideration of sequencing (what do you do first, second, third) and co-activation (what combination of influence bases is required). We frame this cautiously as a plausible educational application derived from the present analysis, not as an intervention that this study directly tested.

## Limitations and future directions

This study used a vignette-based questionnaire with a forced single-choice selection plus a brief written rationale. While this design supports pragmatic access to world-level coaches and elicits situated reasoning, it cannot capture real-time behaviour, emotional arousal, or the multi-party dynamics of in-situ aquatic play. In particular, forcing a single choice likely suppresses co-activation, sequencing and hybridisation of influence bases, which are central to contemporary understandings of coaching as adaptive coordination.

The small, homogeneous sample (three male, world-champion coaches from a highly specific performance culture) provides high information power for the focal phenomenon but also heightens risks of elite exceptionalism and social desirability. Participants may reasonably present their practice in autonomy-supportive terms that align with professional norms, and the vignette format may further encourage ideal-type responses. We therefore interpret the absence of overt conflict or coercive practices as a property of these data, not as evidence that such dynamics do not occur.

Finally, although we used French and Raven's taxonomy and Foucauldian concepts as sensitising frames, our analytic claims remain bounded by what written rationales can reveal. Future studies should triangulate vignette reasoning with in-situ observation, video-stimulated recall and athlete perspectives, and should consider elicitation formats that allow participants to select multiple bases and to narrate temporal sequencing. Comparative work across other low-verbal

or high-risk environments (e.g., apnoea disciplines, combat sports with restricted coaching time, or noisy stadium contexts) would help determine whether the observed compression of influence bases is a broader feature of bandwidth-constrained coaching.

These limitations constrain the strength of knowledge claims. The study speaks to how coaches publicly narrate influence in recurrent scenarios, rather than demonstrating enacted power relations in situ. Accordingly, interpretive statements are bounded to reported practical reasoning and should be read as hypothesis-generating for observational and longitudinal designs that can capture resistance, breakdown, and repair under real competitive pressure.

## Conclusions

This study characterised how world-level underwater rugby coaches reported mobilising interpersonal power across ten recurrent, high-stakes scenarios using a vignette questionnaire with a single forced choice and an open-text rationale. The closed tallies indicated exclusive reliance on referent and expert bases as first-line priorities; informational did not appear as a discrete choice, while legitimate, reward and coercive were absent. The rationales clarified this pattern: coaches described 'silent leadership' (trust, identification, shared rhythm and exemplariness) as the entry point, followed by concise, competence-led guidance, with reason-giving folded into expert delivery. The principal contribution is therefore a context-specific refinement rather than a universal theory claim. In low-verbal aquatic coaching, influence was portrayed as a compressed choreography of trust, demonstration and bounded governance.

## Supporting information

**S1 Dataset. Closed-choice selections by coach across the 10 vignette scenarios.** CSV file containing the anonymised forced-choice selections used to produce the descriptive counts reported in the manuscript.
(CSV)

**S1 Table. Scenario selections.** Word file reporting the selected first-line response option for each vignette scenario by each participating coach.
(DOCX)

**S1 Appendix. Vignette instrument.** Word file containing the full vignette questionnaire, including the 10 scenarios and the response-option structure used for data generation.
(DOCX)

**S1 Analytic Codebook. Reflexive thematic analysis codebook.** Word file containing the analytic code structure, code descriptions, and theme-development framework used during qualitative analysis.
(DOCX)

## Acknowledgments

The authors thank the participating coaches from the National Underwater Sports Federation for their time and insights.

## Author contributions

**Conceptualization:** Samuel Jose Gaviria-Alzate, Claudia Arcila-Rojas.

**Formal analysis:** Samuel Jose Gaviria-Alzate.

**Investigation:** Samuel Jose Gaviria-Alzate.

**Methodology:** Samuel Jose Gaviria-Alzate, Luis Álvarez González, Juliana Posada Monsalve, Jonnathan Arenas Taborda, Lina Restrepo.

**Supervision:** Samuel Jose Gaviria-Alzate, Lina Restrepo.

**Writing – review & editing:** Samuel Jose Gaviria-Alzate, Wilder Geovanny Valencia-Sánchez.

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
