## [Decision Letter · Decision Letter 0]

7 Jan 2026

Dear Dr. Gaviria-Alzate,

Thank you for submitting your manuscript to PLOS ONE. After careful consideration, we feel that it has merit but does not fully meet PLOS ONE’s publication criteria as it currently stands. Therefore, we invite you to submit a revised version of the manuscript that addresses the points raised during the review process.

We look forward to receiving your revised manuscript.

Kind regards,

Jenna Scaramanga

Staff Editor

PLOS One

Journal Requirements:

3. In the online submission form, you indicated that data availability. All supporting materials for this study, including the interview vignette, codebook, reflexive memos, and de-identified excerpts, are available from the corresponding author upon reasonable request. Access will be granted without undue restriction to any interested researcher for academic use, in line with ethics approval and participant confidentiality safeguards.

5. Please amend your authorship list in your manuscript file to include author Wilder Geovanny Valencia-Sánchez..

6. Your abstract cannot contain citations. Please only include citations in the body text of the manuscript, and ensure that they remain in ascending numerical order on first mention.

7. Please include a caption for figure 1.

Reviewers' comments:

Reviewer's Responses to Questions

**Comments to the Author**

1. Is the manuscript technically sound, and do the data support the conclusions?

Reviewer #1: Yes

2. Has the statistical analysis been performed appropriately and rigorously?

Reviewer #1: Yes

3. Have the authors made all data underlying the findings in their manuscript fully available?

Reviewer #1: Yes

4. Is the manuscript presented in an intelligible fashion and written in standard English?

Reviewer #1: Yes

Reviewer #1: Power and influence in world-level sport coaching: A Foucauldian and Raven-Informed phenomenological study

1. Introduction

Key Weaknesses & Needed Improvements

1. Over-theorisation at the expense of problem clarity

The introduction is theoretically dense, integrating Foucault, French and Raven, Bourdieu, Gramsci, and Arendt in rapid succession. While intellectually ambitious, this breadth dilutes conceptual focus. The manuscript does not sufficiently justify why these frameworks must be combined, nor how tensions between them are resolved. The result is a sophisticated but diffuse theoretical backdrop rather than a sharply articulated research problem.

2. Limited articulation of empirical gap

The authors assert that aquatic, low-verbal environments demand different forms of influence, yet do not adequately demonstrate what is missing in existing empirical coaching research. The gap is framed more as a theoretical opportunity than as a clearly evidenced absence in the literature. A more explicit contrast with existing elite coaching studies, especially those already using power or leadership frameworks, would strengthen justification.

3. Aim and contribution are reiterated rather than sharpened

The study aim is restated in multiple places with similar wording, but without increasing specificity. What exactly is new, a new mechanism, a new typology refinement, or a context-specific instantiation, remains underdeveloped. The introduction would benefit from explicit research questions or propositions to anchor the analysis.

2. Methodology

Design and Philosophical Stance

4. Phenomenological claims exceed the data generated

Although framed as existential-hermeneutic phenomenology, data were collected via an online vignette questionnaire rather than interviews or embodied observation. This creates a methodological misalignment: phenomenology prioritises lived experience, yet the study relies on hypothetical responses to predefined scenarios. The authors should either (a) moderate phenomenological claims or (b) justify more convincingly how vignette responses constitute lived experience.

5. Vignette design risks theoretical priming

The scenarios were pre-mapped to French and Raven’s power bases at design time. Despite claims that these were “sensitising concepts,” this structure risks confirmatory bias, subtly steering participants toward theoretically expected responses. This weakens claims of inductive analysis and should be acknowledged more critically.

Participants

6. Extremely small and homogeneous sample

The sample consists of only three male coaches, all world-champion level. While information power is cited as justification, the manuscript nevertheless over-extends interpretive claims relative to sample size. Gender homogeneity, cultural specificity, and elite exceptionalism are insufficiently problematised. Claims about “coaching practice” should be more explicitly bounded to this niche population.

Data Collection and Analysis

7. Forced single-choice selections oversimplify influence processes

Each vignette required a single power-base selection, which artificially constrains co-activation, sequencing, and hybridity, ironically central claims of the paper. This design choice directly contributes to the absence of informational, legitimate, reward, and coercive power in the tallies, undermining the robustness of descriptive findings.

8. Reflexive thematic analysis lacks analytic depth in places

While RTA procedures are correctly cited, theme development remains largely descriptive and confirmatory. The themes mirror the theoretical framing closely, raising concerns that analysis is theory-led rather than data-led. There is limited engagement with ambiguity, contradiction, or deviant cases.

3. Results

Quantitative (Descriptive) Component

9. Frequency tables add limited analytic value

With n = 3, the presentation of frequencies, percentages, and visual heatmaps risks false precision. Although the authors caution against inference, the visual emphasis may mislead readers. The manuscript would be stronger if descriptive tallies were clearly subordinated to qualitative interpretation, or omitted altogether.

Qualitative Themes

10. Themes insufficiently differentiated

The three themes, referent trust, expert demonstration, and calibrated authority, are conceptually overlapping. Distinctions between referent and expert power blur in practice, yet the analysis does not sufficiently interrogate this ambiguity. More analytic tension would strengthen credibility.

11. Lack of counter-narratives or resistance

All accounts portray ethically refined, autonomy-supportive coaching. The absence of conflict, failure, or problematic influence practices suggests social desirability bias, which is acknowledged only briefly. Stronger reflexive interrogation is needed here.

4. Discussion

Theoretical Integration

12. The discussion reiterates rather than extends theory

While theoretically rich, the discussion largely reaffirms existing Foucauldian and leadership concepts instead of advancing them. The claim that informational power is “subsumed” within expert power is plausible but insufficiently theorised, does this suggest a contextual collapse of power bases, or a limitation of the taxonomy?

13. Limited engagement with alternative explanations

Other plausible explanations, elite coach professionalisation, cultural norms in underwater rugby, or shared pedagogical training, are mentioned but not analytically explored. This weakens causal plausibility and theoretical contribution.

5. Practical Implications

14. Implications are generic relative to claims of contextual specificity

Despite strong claims about underwater, low-verbal environments, recommendations for coach education (e.g., trust-building, concise cues, selective authority) are broad and already well-established in coaching literature. More concrete, discipline-specific pedagogical strategies would enhance applied value.

6. Limitations and Future Directions

15. Limitations acknowledged but under-theorised

Although methodological constraints are clearly listed, the implications of these limitations for knowledge claims are not fully unpacked. For instance, how might vignette-based self-report systematically differ from in-situ behaviour under aquatic risk?

**Do you want your identity to be public for this peer review?** For information about this choice, including consent withdrawal, please see our For information about this choice, including consent withdrawal, please see our Privacy Policy .

Reviewer #1: **Yes:** Jet Clayton LongakitJet Clayton Longakit

---

## [Author Response · Author response to Decision Letter 1]

22 Jan 2026

Response to Editor and Reviewers (Revised Submission)

Manuscript title: Power and influence in world-level sport coaching: A Foucauldian and Raven-informed interpretive vignette study in underwater rugby

Corresponding author: Samuel José Gaviria-Alzate

Dear Ms Scaramanga and Editors,

Thank you for the opportunity to revise our manuscript and for the detailed feedback from the staff editor and Reviewer #1. We have made substantive revisions to improve problem clarity, align the methodological framing with the vignette-based design, sharpen the contribution, and strengthen transparency around the descriptive component and data availability.

For ease of review, all changes in the revised manuscript are marked with yellow highlighting and underlining.

A. Staff editor / journal requirements

1. PLOS ONE style requirements and file naming

We checked the manuscript against the PLOS ONE formatting requirements and will follow the journal’s file naming conventions at upload. No content-related changes were required beyond routine formatting consistency.

2. Ethics statement only in the Methods section

We ensured the ethics approval/consent statement appears only within the Methods section and removed any redundant ethics wording elsewhere.

3-4. Data availability and open-data policy (including ‘available on acceptance’)

We revised the Data Availability statement to provide the de-identified closed-choice data underlying Table 3 as Supporting Information (S1 Dataset, CSV) and to provide the vignette prompts/instrument structure (S2 Appendix), analytic codebook (S3 Appendix), and reflexive memo (S4 Appendix). Given the small, readily identifiable elite population, we explain why full open-text rationales cannot be publicly shared without increasing re-identification risk, and we provide an access pathway for qualified researchers to request a further-redacted version under an appropriate data-sharing agreement.

5. Authorship list includes Wilder Geovanny Valencia-Sánchez

We updated the author list to include Wilder Geovanny Valencia-Sánchez.

6. Abstract contains no citations; citations remain in ascending numerical order

We confirmed that the abstract contains no citations. Citations are confined to the body text and retain ascending numerical order on first mention.

7. Include a caption for Figure 1

We included a caption for Figure 1 and updated Figure 1 to a conceptual model aligned with the thematic findings.

8. Reviewer-suggested citations

We evaluated the reviewer’s suggestions and incorporated additional citations only where they directly strengthened the empirical gap and interpretation; we did not add citations that were not materially relevant to the argument.

B. Response to Reviewer #1 (Jet Clayton Longakit)

We reproduce each comment in condensed form and respond point-by-point below. Locations refer to the revised manuscript sections; all edits are highlighted/underlined in yellow.

1. Over-theorisation at the expense of problem clarity

We streamlined the Introduction to prioritise the empirical problem (bandwidth-constrained influence in underwater rugby) and the rationale for the two primary lenses (Foucauldian power/knowledge and French-Raven power bases). Additional theoretical signposts (e.g., Bourdieu/Arendt) are now used sparingly to extend interpretation rather than to frame the problem statement. See Introduction; Discussion (theoretical integration).

2. Limited articulation of empirical gap

We strengthened the empirical gap by more explicitly contrasting the underwater, low-verbal context with mainstream elite coaching studies and by specifying what is under-described in the literature: the mechanisms and sequencing of influence when verbal bandwidth is structurally constrained. See Introduction (problem statement and gap paragraphs).

3. Aim and contribution reiterated rather than sharpened

We tightened the study aim and contribution and presented explicit research questions to anchor the analysis. See end of Introduction (research questions).

4. Phenomenological claims exceed the data generated

We aligned the methodological framing with the actual data source by presenting the study as an interpretive qualitative vignette study with a hermeneutic sensibility, rather than as phenomenology based on lived-experience interviews/observation. We moderated any claims that implied direct access to lived experience and clarified what vignette reasoning can and cannot support. See Methods (Design and philosophical stance) and Limitations.

5. Vignette design risks theoretical priming

We expanded our critical acknowledgement of priming risk from the design-time mapping of options to power bases and clarified how we mitigated this risk: power terminology was not shown to participants; rationales were analysed inductively; and the taxonomy was used as sensitising concepts at interpretation. See Methods (Scenario design) and Limitations.

6. Extremely small and homogeneous sample

We further bounded our claims to this niche, elite sample and strengthened the rationale using information power and context-specific aims. We also foreground the implications of gender/cultural specificity as boundary conditions. See Participants; Limitations and Future Directions.

7. Forced single-choice selections oversimplify influence processes

We clarified that the forced single-choice design compresses co-activation and sequencing and therefore cannot be interpreted as prevalence or ‘effectiveness’. To improve transparency, we (i) retained only descriptive counts (Table 3), (ii) provide scenario-by-scenario selections as S1 Table, and (iii) provide the de-identified closed-choice dataset as S1 Dataset. See Methods (Descriptive depiction); Results; Supporting Information.

8. Reflexive thematic analysis lacks analytic depth in places

We strengthened the analytic narrative by sharpening theme boundaries, adding explicit engagement with ambiguity and plausible alternative readings, and expanding reflexive commentary on social desirability and elite self-presentation. See Results (themes) and Discussion (alternative explanations).

9. Frequency tables/visuals add limited analytic value; risk false precision

We removed the heatmap visual emphasis and retained only a compact frequency table (Table 3) as orientation, explicitly subordinated to the qualitative interpretation. Scenario-level detail is placed in S1 Table, and we reiterate that counts are descriptive only with n=3. See Results (Table 3 note; accompanying text) and Supporting Information.

10. Themes insufficiently differentiated

We revised theme descriptions to make the analytic distinctions explicit (relational anchoring vs. expertise-as-demonstration vs. bounded governance), while also acknowledging co-activation and sequencing as an empirical finding rather than a thematic overlap error. See Results (theme introductions and transitions) and Figure 1 (conceptual model).

11. Lack of counter-narratives or resistance

We expanded the reflexive account of why ‘polished’ narratives may appear in vignette self-report and explicitly cautioned readers against equating absence of conflict in these data with absence of conflict in practice. See Results (reflexive paragraph) and Limitations.

12. Discussion reiterates rather than extends theory; ‘subsumed’ informational power needs clearer theorisation

We elaborated the theoretical contribution by interpreting the pattern as a contextual compression/hybridisation of power bases under low-verbal constraints, rather than as a simple absence of informational influence. See Discussion (theoretical integration; taxonomy refinement).

13. Limited engagement with alternative explanations

We expanded the Discussion to more fully consider alternative explanations (elite coach professionalisation, shared pedagogical training, cultural norms and selection effects) and to clarify how these explanations intersect with the underwater ecology account. See Discussion (Alternative explanations and boundary conditions).

14. Practical implications are generic relative to contextual specificity

We revised the Practical implications to include concrete, underwater-specific pedagogical strategies (e.g., minimal cue lexicons, standardised hand/tactile signals, breath-window ‘micro-scripts’, and safety governance protocols). See Practical implications.

15. Limitations acknowledged but under-theorised

We deepened the limitations by explaining how vignette-based self-report may systematically differ from in-situ behaviour under aquatic risk and by specifying how the forced-choice format may suppress co-activation and temporal sequencing. See Limitations and Future Directions.

C. Supporting Information files provided with this revision

S1 Dataset (CSV): De-identified closed-choice selections underlying Table 3.

S1 Table: Scenario-by-scenario closed-choice selections (descriptive).

S2 Appendix: Vignette prompts and decision-task structure (English translation and Spanish original).

S3 Appendix: Analytic codebook.

S4 Appendix: Reflexive memo documenting analytic decisions.

Sincerely,

The authors

---

## [Decision Letter · Decision Letter 1]

9 Mar 2026

Dear Dr. Gaviria-Alzate,

Thank you for submitting your manuscript to PLOS ONE. After careful consideration, we feel that it has merit but does not fully meet PLOS ONE’s publication criteria as it currently stands. Therefore, we invite you to submit a revised version of the manuscript that addresses the points raised during the review process.

We look forward to receiving your revised manuscript.

Kind regards,

Ender Senel, PhD

Academic Editor

PLOS One

**Journal Requirements:**

Reviewers' comments:

Reviewer's Responses to Questions

**Comments to the Author**

Reviewer #1: All comments have been addressed

Reviewer #2: (No Response)

2. Is the manuscript technically sound, and do the data support the conclusions?

Reviewer #1: Partly

Reviewer #2: (No Response)

3. Has the statistical analysis been performed appropriately and rigorously?

Reviewer #1: Yes

Reviewer #2: Yes

4. Have the authors made all data underlying the findings in their manuscript fully available?

Reviewer #1: Yes

Reviewer #2: Yes

5. Is the manuscript presented in an intelligible fashion and written in standard English?

Reviewer #1: Yes

Reviewer #2: Yes

Reviewer #1: The revised manuscript demonstrates serious scholarly engagement, improved methodological integrity, and a well-bounded interpretive contribution. While the theoretical advance remains cautious rather than transformative, this is appropriate given the design and sample.

Reviewer #2: The rationale for selecting the theoretical frameworks could be clarified further.

Using a single-option data collection format would be beneficial. The rationale for the sample could be explained more clearly.

Given the limited number of participants, it may be helpful to address how data saturation was taken into account.

Further detail on the scenario-based approach would improve methodological transparency (validity and reliability).

The analytic process could be described in greater detail, particularly regarding how codes were generated and developed into themes.

Including additional illustrative quotations from participants could help demonstrate how themes were derived from the data.

Inter-rater reliability was not pursued. It would be helpful to clarify whether the analysis conducted was considered sufficient and how analytic rigor was ensured within this process.

The discussion could further explore why referent power and expert power emerged as the most prominent bases.

The connection between the theoretical perspective and the analysis could be articulated more clearly.

Language editing is recommended to improve clarity.

**Do you want your identity to be public for this peer review?** For information about this choice, including consent withdrawal, please see our For information about this choice, including consent withdrawal, please see our Privacy Policy .

Reviewer #1: **Yes:** Jet LongakitJet Longakit

Reviewer #2: No

---

## [Author Response · Author response to Decision Letter 2]

10 Mar 2026

Dear Academic Editor and Reviewers,

We sincerely thank you for the careful reading of our revised manuscript and for the constructive observations that helped us sharpen the contribution, clarify the methodological logic, and better bound our claims. We have revised the manuscript accordingly. In the marked-up manuscript, the changes introduced in this round are shown in blue.

Academic Editor / overall remaining issues

1. Theoretical contribution remains incremental rather than decisive.

Response: We agreed that the manuscript needed a more explicit and tighter statement of what kind of theoretical move is being made. We therefore revised the end of the Introduction, the opening of the Discussion, and the Conclusion to clarify that the contribution is not presented as a universal revision of French and Raven’s taxonomy, nor as a general pedagogical model of elite coaching. Instead, we now state more directly that the manuscript advances a context-specific refinement for bandwidth-constrained coaching ecologies. Referent and expert power emerge as the dominant actionable pair, informational influence is often nested within expert guidance, and legitimate authority remains a bounded resource for safety and standards.

2. Closed-choice data still add limited analytic leverage.

Response: We appreciated this point and clarified the role of the closed-choice data in three places: in the Methodology subsection 'Descriptive depiction of selections', in the opening paragraph of the Results section presenting Table 3, and in 'Methodological contribution'. We now state more explicitly that the tallies are analytically modest on their own but were retained because they preserve what each coach foregrounded when compelled to prioritise a single first-line action. Read alongside the rationales, they provide an orienting layer that helps identify convergence, qualification, and hidden co-activation.

3. Conceptual model (Figure 1) risks reification.

Response: We revised the note to Figure 1 to state explicitly that the model is heuristic rather than sequential, exhaustive, or prescriptive. The revised note now frames the figure as a synthesis of the themes under low-verbal aquatic conditions rather than a fixed process model.

4. Scope of implications could still be more tightly bounded.

Response: We tightened the first paragraph of 'Practical implications for coach education' and the closing paragraph of that section. The revised wording now distinguishes between direct implications for underwater rugby and comparable low-verbal, risk-aware environments, and broader relevance framed only as a transferable hypothesis for future comparative work rather than a general prescription.

Reviewer #2

1. The rationale for selecting the theoretical frameworks could be clarified further.

Response: We revised both the Introduction and 'Design and philosophical stance' to explain more clearly why Foucauldian concepts and French and Raven’s taxonomy were combined. We now state that the frameworks were selected because they operate at complementary analytic scales. Foucauldian concepts help interpret the broader governance and normalisation of conduct, whereas French and Raven’s framework offers a mid-range vocabulary for the interactional influence levers visible in concrete vignette responses.

2. Using a single-optiona data collection format would be beneficial. The rationale for the sample could be explained more clearly.

Response: We clarified both points. First, we standardised the manuscript language to reflect that each vignette required a single forced-choice response, and we added a methodological justification explaining that this design was used to compel prioritisation of the action coaches would most likely take first under time-pressured conditions. Second, in 'Participants' we expanded the sampling rationale and clarified that this was a deliberately narrow, information-rich elite sample chosen for analytic adequacy rather than representativeness.

3. Given the limited number of participants, it may be helpful to address how data saturation was taken into account.

Response: We addressed this directly in the 'Participants' subsection. We now state that saturation was not pursued as a formal endpoint because the study did not aim at broad code saturation across a heterogeneous field. Instead, adequacy was judged in relation to the narrow research question, the rarity of the population, and the density of elicitation generated by ten scenarios per participant.

4. Further detail on the scenario-based approach would improve methodological transparency (validity and reliability).

Response: We expanded 'Scenario design' and the questionnaire description to clarify the instrument logic. The revised text now explains the purpose of the design-time mapping, the rationale for the single-option format, and the fact that the vignette guide functioned as an elicitation device rather than as a psychometric scale. We also clarified that methodological attention was directed to contextual realism, face validity, and discriminability of options through expert review, and that the full vignette wording and response structure are supplied in S2 Appendix.

5. The analytic process could be described in greater detail, particularly regarding how codes were generated and developed into themes.

Response: We substantially expanded 'Reflexive thematic analysis (RTA)' to describe the analytic sequence more explicitly. The manuscript now outlines six recursive steps: familiarisation, open coding, memo-writing, clustering of related codes, iterative development and renaming of themes, and a final interpretive pass relating themes back to the descriptive selections and the sensitising frameworks. We also added examples of the kinds of action-oriented codes that were grouped into broader thematic patterns.

6. Including additional illustrative quotations from participants could help demonstrate how themes were derived from the data.

Response: We carefully considered this suggestion. Because the written rationales were intentionally brief and the sample is a very small, deductively identifiable elite population, we remained cautious about adding numerous standalone quotations that might add limited interpretive depth while increasing disclosure risk. Instead, we strengthened excerpt-to-theme traceability by clarifying the reporting logic in Methods and by linking each theme more explicitly to the scenario clusters and participant-specific patterns that generated it. We believe this preserves interpretive transparency while remaining proportionate to the nature of the dataset.

7. Inter-rater reliability was not pursued. It would be helpful to clarify whether the analysis conducted was considered sufficient and how analytic rigor was ensured within this process.

Response: We revised 'Design and philosophical stance' to state more explicitly that, consistent with reflexive thematic analysis, rigour was not sought through coder agreement. We now clarify that analytic rigour was pursued through reflexivity, transparency, memoing, an audit trail, and structured critical-friend interrogation of theme boundaries and evidential adequacy.

8. The discussion could further explore why referent power and expert power emerged as the most prominent bases.

Response: We revised the Discussion to elaborate this point more clearly. The new text explains that referent and expert power are especially actionable under low-verbal, time-compressed, breath-constrained conditions. Trust and identification help secure rapid receptivity, while demonstrated competence carries the actionable adjustment. We also added a sentence clarifying why reward, coercive, and even legitimate strategies may be less functional as first-line responses in this particular ecology.

9. The connection between the theoretical perspective and the analysis could be articulated more clearly.

Response: We strengthened this connection in the Introduction, Methodology, and Discussion. The revised text now more explicitly links Foucauldian concepts to the governance of conduct and French and Raven’s taxonomy to the interactional prioritisation visible in the scenarios and then reconnects both levels in the Discussion when interpreting the compressed choreography of trust, demonstration, informational guidance, and bounded authority.

10. Language editing is recommended to improve clarity.

Response: We carefully edited the manuscript throughout for clarity, consistency, and precision. This included tightening the framing of the contribution, standardising the description of the single forced-choice design, removing residual wording that implied interviews, and smoothing several passages in the Methods, Results, Discussion, and Conclusion.

We hope that these revisions have addressed the remaining concerns and improved the manuscript’s clarity, methodological coherence, and contribution. We are grateful for the opportunity to revise the paper further.

Sincerely,

Samuel José Gaviria Alzate and co-authors

---

## [Editor Report · Decision Letter 2]

11 Mar 2026

Power and influence in world-level sport coaching: A Foucauldian and Raven-informed interpretive vignette study in underwater rugby

PONE-D-25-60644R2

Dear Dr. Gaviria-Alzate,

We’re pleased to inform you that your manuscript has been judged scientifically suitable for publication and will be formally accepted for publication once it meets all outstanding technical requirements.

Kind regards,

Ender Senel, PhD

Academic Editor

PLOS One

---

## [Editor Report · Acceptance letter]

PONE-D-25-60644R2

PLOS One

Dear Dr. Gaviria-Alzate,

I'm pleased to inform you that your manuscript has been deemed suitable for publication in PLOS One. Congratulations! Your manuscript is now being handed over to our production team.

Kind regards,

on behalf of

Dr. Ender Senel

Academic Editor

PLOS One